# Sequential Injection Analysis for Automation and Evaluation of Drug Liberation Profiles: Clotrimazole Liberation Monitoring

**DOI:** 10.3390/molecules26185538

**Published:** 2021-09-12

**Authors:** Hana Sklenářová, Marek Beran, Lucie Novosvětská, Daniela Šmejkalová, Petr Solich

**Affiliations:** 1Department of Analytical Chemistry, Faculty of Pharmacy in Hradec Králové, Charles University, 500 05 Hradec Králové, Czech Republic; beranma1@faf.cuni.cz (M.B.); lucie.novosvetska@gmail.com (L.N.); solich@faf.cuni.cz (P.S.); 2Contipro a.s., 561 02 Dolní Dobrouč, Czech Republic; daniela.smejkalova@contipro.com

**Keywords:** clotrimazole, Franz cell, kinetic profile, liberation study, sequential injection analysis

## Abstract

A fully automated sequential injection system was tested in terms of its application in liberation testing, and capabilities and limitations were discussed for clotrimazole liberation from three semisolid formulations. An evaluation based on kinetic profiles obtained in short and longer sampling intervals and steady-state flux values were applied as traditional methods. The obtained clotrimazole liberation profile was faster in the case of Delcore and slower for Clotrimazol AL and Canesten cream commercial formulations. The steady-state flux values for the tested formulations were 52 µg cm^−2^ h^−1^ for Canesten, 35 µg cm^−2^ h^−1^ for Clotrimazol AL, and 7.2 µg cm^−2^ h^−1^ for Delcore measured in 4 min sampling intervals. A simplified approach for the evaluation of the initial rate based on the gradient between the second and third sampling points was used for the first time and was found to correspond well with the results of the conventional methods. A comparison based on the ratio of the steady-state flux and the initial rate values for Canesten and Clotrimazol AL proved the similarity of the obtained results. The proposed alternative was successfully implemented for the comparison of short-term kinetic profiles. Consequently, a faster and simpler approach for dissolution/liberation testing can be used.

## 1. Introduction

Drug liberation is a critical initial step in the sequence of how an active substance reaches the target location in the body. It describes the process in which the active ingredient from the administered semisolid drug formulation based on gel or ointment is liberated and becomes available to pass the skin barrier. The liberation kinetics is therefore an important characteristic of each pharmaceutical as it affects the pharmacokinetics, which needs a thorough evaluation. A wide variety of methodologies related to the many formulation types have been used to determine the liberation kinetics [1]. Obviously, methods are required to enable the rapid, reliable, and reproducible evaluation of drug liberation, and further improvement of the available methods is needed.

Applications of flow methodologies in monitoring studies have been proven valuable as fully automated alternatives to offline dissolution and liberation tests [1]. Methods based on flow injection analysis (FIA) and sequential injection analysis (SIA) manifold have been applied, varying in the type of drug formulations (tablets, capsules, etc.), parameters (dissolution or liberation), and active substance [2,3,4,5,6,7,8,9,10]. Among the different modifications, an important step was introducing more liberation units in one analyzer system, resulting in little time-lapse, evaluation of repeatability, and result confirmation [2,3,5]. Recently, we reported the implementation of low-pressure separation to monitor more than one active substance in a single formulation [6]. Flow analyzers were also proven to be compatible with the liberation tests of semisolid samples (topical formulations) through porcine skin or different synthetic membranes [4,5,10]. In addition, a flow system that combines dissolution with permeation testing was proposed [9]. An excellent comprehensive review focused on automation in pharmaceutical analysis up until 2006, including liberation testing [8].

Compared to offline sampling from liberation units, fully automated sampling enables real-time observations of the detailed kinetic profile. Offline testing can suffer from manual sampling errors and a risk of decompositions of the active substance while the offline taken sample is stored before the measurement. Pharmacopeial requirements are not based on detailed kinetic profiles, but on the limits of the liberated substance content after the respective time that should be attained/exceeded. Although new and reference formulations in frames of bioequivalence studies are compared, detailed kinetic profiles should be observed. In case of commercial dissolution systems [11], there is the possibility of sampling online when direct UV spectrophotometry is used for active substance determination. In such systems, short sampling intervals can be used too. The other way of automation is based on online sampling when samples are stored in the autosampler of an HPLC instrument and then analyzed at-line (not in real-time).

As Franz diffusion cells were used for liberation testing, the same units with different dimensions [12] can be applied not only for liberation testing, but also for permeation testing or other interaction studies where different artificial or model membranes and even inserts with cultivated cell lines [13] are placed in the Franz cell design. In the case of liberation, different technological parameters in pharmaceutical formulation development can be observed, e.g., enrofloxacin incorporation into liposomes [14], cannabidiol release through rabbit ear skin [15], and intercalation of diclofenac and naproxen to slow down their release [16]. The bioequivalence of new permethrin cream formulation [17] was tested using Franz cells. The other applications are based on the permeability testing of antiretroviral drugs, lamivudine and zidovudine, bioavailability [18], or a Franz cell biosensor for hydrogen peroxide penetration through skin and dialysis membranes [19].

In the current work, we demonstrate the advantages of test automation by SIA for fast monitoring of clotrimazole liberation. Clotrimazole (structure shown in Figure 1 [20]) is an antifungal drug that can be taken orally as a pill or applied to the skin as a cream [21]. This antimycotic agent was studied previously to achieve improvements in modern formulation processes. For example, tablets with different excipients were examined to modify dissolution profiles using chitosan and thioglycolic acid conjugates [22]. Improvements also included the use of micro- and nanoemulsions [23], nanocapsules [24], and the release from mucoadhesive clotrimazole-loaded nanofibers [25]. The problem of the low solubility of clotrimazole in aqueous buffers has been discussed concerning the effect of temperature [26], unusual clotrimazole dissolution rate affected by supersaturation in microbiological growth medium [27], and in vitro release of clotrimazole from mucoadhesive thermoreversible gels [28].

A general shortcoming of offline sampling from liberation units is the lower number of sampling points preventing detailed kinetic profiles from being obtained in such tests. The other challenge is based on the delay between sampling and analysis where decomposition of these active substances can occur, and manual sampling may suffer from human errors. In the evaluation of quick liberation profiles, similar to clotrimazole, the first sampling points can substantially affect the liberation profile; thus, quick sampling provides more information mainly in this first part of the liberation test.

This work describes the limits of simple online determination using the direct detection of clotrimazole. The effect of the sampling period length is discussed, with a critical comparison of these profiles obtained with different sampling intervals. We propose a simple method to calculate the initial liberation rate from the second and third sampling points of the registered profile to characterize the release of clotrimazole from three semisolid formulations. This novel approach is compared with the values obtained by the standard evaluation methods, and we discuss the method as an alternative to obtain the required information in shorter times, using an example of different clotrimazole formulations with an expected quick release of the active substance.

## 2. Results and Discussion

### 2.1. Limits of Liberation Automation

The limits of the SIA system in terms of flow rate, sampling volume, number of samples taken in one point, time for overall procedure for the sampling step including liberation medium refilling, and determination sensitivity were studied. The online connection of three Franz cells (liberation units) to one SIA system caused longer sampling periods and delayed sampling from the respective Franz cell. In the case of three units, a three-times-longer sampling interval was a compromise to obtain a detailed kinetic profile without affecting the obtained curve. This was tested by comparison with one Franz cell experiment with short sampling interval, while possibilities and limitations were identified.

To study the differences in the first part of the liberation profile, fast liberation of clotrimazole was selected. The limitations of online determination in terms of sensitivity, repeatability, and analysis time were the main parameters in the study considered in terms of the expected liberation demands. The conditions of the liberation test were in accordance with the European Pharmacopoeia [1], including temperature, Franz cell geometry, liberation medium, and the minimal number of evaluated liberations (six tests). The evaluation was then extended to describe other possibilities enabled by the detailed kinetic profile.

### 2.2. Optimization of Clotrimazole Determination

Clotrimazole determination was optimized with respect to the determination sensitivity, while following our previous experience 5–10% of the expected contents in the pharmaceutical formulation should be detected. The solubility of clotrimazole in water is limited due to the lipophilic characteristic of the substance (Figure 1). Therefore, we worked with its solutions in both aqueous buffer and ethanol. Calibrations in these two media were compared to confirm liberation into the phosphate buffer that was not controlled by clotrimazole solubility.

Calibration was tested in the range of 2.5–20 mg L^−1^ including seven calibration points where the average of measurements in triplicate was used. The linearity was calculated with a determination coefficient of 0.9976. The limited solubility in phosphate buffer affected the measurements at concentrations exceeding 30 mg L^−1^. The limit of detection (LOD; 0.8 mg L^−1^, 3% of the expected clotrimazole level) and limit of quantitation (LOQ; 2.2 mg L^−1^, 8%) were calculated as concentrations, producing a 3- and 10-times higher signal compared to the signal of noise, respectively [29].

### 2.3. Optimization of Liberation Test

A sample volume of 50 µL was selected as a compromise between sensitivity and sample consumption in the liberation test, which should be kept low to avoid affecting the liberation profile. The flow rate was optimized for detection to ensure sufficient sensitivity while not extending the determination time, since the sampling periods should be short to obtain a detailed kinetic profile. This parameter is important in the case of multiple liberation monitoring. We used three Franz cells and a single flow system. This setup corresponded to three-times-longer sampling periods. The optimal flow rate for the detection step was 10 µL s^−1^.

The other parameters, such as flow rates in other analysis steps, did not affect the sensitivity but only the analysis time. To ensure a short analysis time, a higher flow rate of 50 µL s^−1^ was used. Sampling was carried out twice to decrease the signal noise affecting the precision of clotrimazole determination at the respective sampling interval.

### 2.4. Comparison of Kinetic Profiles

We compared the kinetic profiles monitored in three Franz cells using longer sampling intervals with those obtained in single cell experiment to confirm that the profiles were not affected by the sampling periods. The kinetic profiles of the clotrimazole formulation obtained with shorter and longer sampling intervals are displayed in Figure 2. The profiles for all formulations proved that longer periods did not affect the shape of the plot. However, it is clearly visible that the liberation of clotrimazole is faster than if derived from three Franz cells in parallel. The steady state was reached after 25 min for Canesten, 30 min for Clotrimazol AL, and 4 min for Delcore.

The quantity of liberated clotrimazole was recalculated with respect to dilution and the weighted amount of the respective formulation. The expected concentration corresponding to the clotrimazole content in the tested creams was 27 mg L^−1^. In our experiments, a clotrimazole concentration close to 30 mg L^−1^ was released from Clotrimazol AL and about 20 mg L^−1^ from Canesten. The content of clotrimazole in the Delcore formulation was higher but a smaller amount was applied for the test to obtain a comparable content. The overall clotrimazole amount reached 6 mg L^−1^, while our test detected 4 mg L^−1^.

The kinetic profiles of all three tested formulations were then compared using longer sampling intervals in three Franz cells repeated twice, and produced six profiles. The average values are displayed in Figure 3. The results are reported as the percentage of liberated clotrimazole for easier comparison and to include all three formulations. Differences between formulations in the first part of the profiles were observed, whereas fast release of clotrimazole was observed for Delcore, followed by Clotrimazol AL, and the slowest release was from Canesten cream. The final concentration was reached in 36 min for all tested formulations.

Figure 4 demonstrates a comparison of the overall amount of liberated clotrimazole linearized using Equation (2) for Canesten and Clotrimazol AL formulations when the applied cream amount and the overall content of clotrimazole were at the same levels. The release rates were very similar, with a higher release of clotrimazole from Canesten than from Clotrimazol AL. Additionally, the amount of released clotrimazole was higher for Canesten cream. The plots in Figure 4 were obtained from liberations determined in a single Franz cell, which means at short sampling intervals. The linearized profiles of clotrimazole release for all formulations using three Franz cells experiments are presented in the Appendix A.

The steady-state flux values for tested formulations were 52, 35, and 7.2 µg cm^−2^ h^−1^ for Canesten, Clotrimazol AL, and Delcore, respectively, in case of a single Franz cell experiments. These values can be compared with those obtained from measurements using three Franz cells: 54, 38, and 3.6 µg cm^−2^ h^−1^, respectively. The larger difference found for Delcore may have resulted from the rapid clotrimazole release. The flux in our study is lower when compared to the previously reported value of 167–249 µg cm^−2^ h^−1^ measured with a clotrimazole transdermal spray formulation containing skin permeation enhancers [30].

### 2.5. Initial Rate

The initial liberation rate was determined to quantify the release of clotrimazole when our novel approach using the initial parts of the kinetic profiles was applied. Only Canesten and Clotrimazol AL could be compared in terms of total clotrimazole content monitored in one- and three-unit experiments. Higher slope values were observed for Canesten (1.7 and 0.8) than for Clotrimazol AL (0.9 and 0.2). Higher values in the single-unit experiments corresponded to fast release, and thus short sampling intervals more accurately covered the liberation profile shape.

To compare all three formulations, the percentage of liberated clotrimazole was evaluated, and, again, the slope of the line between the second and the third sampling points was considered. The highest percentage values were demonstrated for Delcore (95% and 96%, after 8 and 24 min, respectively), lower values for Clotrimazol AL (70% and 86% for 8 and 24 min, respectively), and Canesten (64% and 87% for 8 and 24 min, respectively). The slopes corresponding to the trend in the liberation after the beginning of the liberation showed a steady state for Delcore (0.3 and 0.2) even in such a short interval after the liberation test began, a slower increase for Clotrimazol AL (5.9 and 1.1), and a continuing increase for Canesten (7.4 and 4.5). The plots are presented in Appendix A.

The ratio of these steady-state flux values for Canesten and Clotrimazol AL was compared with the ratio of the initial rate values. A flux ratio of 1.5 for single-cell and 1.4 for three-cell experiments was found in for the steady-state measurement. The ratio of the initial rates was 1.8 for single-cell and 3.6 for three-cell experiments. Clearly, higher values were observed for the ratio of the initial rates. However, the results revealed a similar trend and can be used for comparison of quickly liberated active substances.

## 3. Materials and Methods

### 3.1. Chemicals and Solutions

Clotrimazole standard (99–101%), potassium chloride (≥99.0%), sodium hydrogen phosphate (99–101%), potassium dihydrogen phosphate (≥99.0%), and ethanol were supplied by Sigma-Aldrich (Prague, Czech Republic). Sodium chloride was purchased from Penta (Prague, Czech Republic). Ultrapure water prepared in a Millipore Milli-Q system (Merck, Prague, Czech Republic) was used for the preparation of all aqueous solutions.

A clotrimazole stock solution was prepared in ethanol at a concentration of 2 mg mL^−1^ and the calibration solutions were obtained by their appropriate dilution with a phosphate buffer (0.1 mol L^−1^, pH = 7.4). The same buffer was applied as the medium in the liberation test [1] with a hydrophilic polycarbonate membrane filter with a 0.4 µm pore size, 37 mm diameter, and thickness of 10 µm (Isopore, Merck, Prague, Czech Republic) [4].

To compare liberation profiles, three formulations were tested: Clotrimazol AL 1% cream (Aliud Pharma GmbH, Laichingen, Germany), Canesten cream 1% clotrimazole (Kern Pharma S.L., Spain), and Delcore, a newly developed 3.3% clotrimazole formulation in the form of lyophilizate for dispersion preparation at the time of use (Contipro a.s., Dolní Dobrouč, Czech Republic). The amount of clotrimazole creams used in the liberation test was 40 mg. A Delcore dispersion of 300 µL was taken from 1 mL phosphate buffer, in which 5 mg of lyophilizate was dissolved, following the producer’s recommendation at the time prior to liberation test. These formulations’ compositions are described in the Appendix A.

### 3.2. Apparatus

The apparatus for automation of the liberation tests and the tubing configuration are shown in Figure 5, which are based on a MicroSIA system (FIAlab^®^ instruments, Seattle, WA, USA). The miniaturized flow manifold comprised a 5 mL automatic piston pump, a holding coil with an inner volume of 400 µL, and a 6-port selection valve. Fluorinated ethylene propylene 1520 XL tubing with 0.75 mm i.d. were used for all flow conduits. A homemade poly(methyl methacrylate) water bath was used to accommodate up to nine Franz cells. In this study, three one-wall Franz cells made of glass, with an inner volume of 15 mL, were placed in a thermostated bath and connected to a single SIA system in parallel. A thermostat (Julabo Labortechnik GmbH, Seelbach, Germany) maintained the Franz cells at a temperature of 32 °C. The water bath was placed onto a 15-position magnetic stirrer (Variomag Poly 15, ThermoFisher, Pardubice, Czech Republic). Stirring bars 10 mm in length were placed inside the Franz cells to mix their contents at 450 rpm.

An additional peristaltic pump (Minipuls 3, Gilson^®^, Middleton, WI, USA) was used to continuously circulate the liberation medium with the respective content of liberated clotrimazole in the sampling loops. Each sampling loop was 130 cm long, made from 90 cm long FEP 1520 XL circle (0.75 mm i.d.), and included a 40 cm long peristaltic pump tube (1.02 mm i.d., Gilson^®^, Watrex, Prague, Czech Republic). The sampling loops connected the Franz cell to one of the ports of the selection valve via a T-connector (Tee Assembly Tefzel^®^, 0.50 mm i.d., Upchurch, Watrex, Prague, Czech Republic) for low dead-volume sampling. The peristaltic pump produced a flow rate of 1.3 mL min^−1^. A spectrophotometric detector (USB4000, FIAlab^®^ instruments, Seattle, WA, USA) with a UV light source (D-2000, Ocean Optics, OptiXs, Prague, Czech Republic) and a 10 mm PTFE flow Z cell was used for detection at an integration time of 50 ms. Optical fibers (600 µm i.d., Ocean Optics, OptiXs, Prague, Czech Republic) connected the flow cell and the detector. Absorbance was monitored at 210 nm, while 500 nm was applied as a reference wavelength. The flow system was controlled by flow programming aided by FIAlab^®^ software for Windows 5.0 (FIAlab^®^ instruments, Seattle, WA, USA).

The material of the membrane used for the liberation tests was carefully selected not to affect or limit the liberation of the active compound from the pharmaceutical formulation. In an earlier study [4], we confirmed that polycarbonate with a pore size of 0.4 µm was the best membrane for studying indomethacin liberation. Our preliminary optimization indicated no limitation of the clotrimazole test.

### 3.3. Measurement Procedure

Initially, the flow system was filled with phosphate buffer, used as the carrier and liberation medium, and each port of the selection valve was washed with the solution of the connected reservoir, e.g., clotrimazole standard or buffer. For measurement, the syringe pump was partly filled with buffer and a 50 µL sample was aspirated from the selection valve into the holding coil and then pushed toward the detection flow cell at a flow rate of 10 µL s^−1^ by the aid of the carrier buffer.

At the beginning of the liberation test, the Franz cells were filled with the buffer/medium and allowed to circulate in the sampling loops for 20 min to remove potentially contained bubbles. Typically, 15 mL was enough to fill the cell and the sampling loops. The bath temperature was kept at 32 °C, being the standard temperature of human skin used in liberation studies [1]. Afterward, the membrane was carefully placed in the Franz cell to stay in contact with the inside liquid and to avoid bubbles trapping beneath the membrane. Then, the desired quantity of pharmaceutical formulation was applied across the overall area of the membrane. The upper part of the Franz cell was fixed by rubber bands, and the open top was covered with a parafilm^®^ (Sigma-Aldrich, Prague, Czech Republic) layer to prevent evaporation.

Before each sampling step, the respective port of the selection valve with a T-connector was washed by the aspiration of 100 µL buffer that was then discharged to waste. A volume of 50 µL was sampled and subjected to the quantification of the contained clotrimazole. Sampling and determination were repeated twice for averaging to compensate for arbitrary errors. Afterward, the buffer volume in the respective sampling loop and Franz cell was replenished to keep the liquid in steady contact with the membrane by pushing 320 µL into the sample loop. The consequent dilution was compensated mathematically during posterior data evaluation.

The entire measurement cycle, including washing the detection cell, took 4 min in for one Franz cell, while sampling from three Franz cells lasted 12 min. The intervals between sampling were kept constant for the complete monitoring period of 48 min with 12 sampling intervals for the single-cell experiment and 108 min in 9 sampling intervals for the three-cell experiment.

### 3.4. Optimization and Data Evaluation

Optimization of the sample volume and the flow rate during detection was carried out with respect to the sensitivity of clotrimazole determination. The liberation test was optimized to keep the sample volume small and the analysis time short enough while ensuring a detailed kinetic profile could still be obtained.

The evaluation of these results obtained in the liberation test was based on recalculation of the clotrimazole concentration at the respective sampling times, taking as a base the calibration with standard solutions measured on the same day. Recalculation was needed to compensate for the dilutions of the liberation medium in the Franz cell and in the sampling loop together with the volume of the acceptor by fresh medium after each sampling point. This recalculation, described elsewhere [5], uses the following Equation (1), where *C_n_* is the concentration of the *n* sample [31]:C _n, corrected_ = C _n, measured_ + volume _sample_/volume _acceptor_ × C _n−1, measured_(1)

### 3.5. Comparison of Kinetic Profiles

Semisolid pharmaceutical formulations, such as Clotrimazol AL, Canesten, and Delcore, were used to compare clotrimazole liberation in detailed kinetic profiles. Each formulation was tested in three Franz cells in parallel. The liberation test was completed with single-cell testing to prove that the liberation kinetics were monitored properly. The average values expressed as a percentage of liberated clotrimazole concentration in six profiles [1] were compared. The overall amount of liberated clotrimazole was evaluated using the following equation:Q_n_ = C _n, corrected_ × volume _acceptor_/diffusion area(2)
where Q_n_ is the cumulative drug amount permeated at each sampling time, related to the area of the polycarbonate membrane exposed to the acceptor liquid [5]. Data were linearized using the square root of time transformation and linear plots were obtained by plotting the cumulative amounts released in µg cm^−2^ per square root of hours (h^1/2^). Additionally, the initial rate of all profiles was determined to obtain a quick correlation. Those results were compared with previously found results using different methods mentioned in the Introduction. The initial rate is expressed as a slope of the line between the second and the third sampling points.

The detailed comparison was carried out for Clotrimazol AL and Canesten cream formulations as these two formulations contain the same clotrimazole level. For approximate comparison of all tested formulations, the percentage of liberated clotrimazole was applied to identify the similarities and trends in the kinetic profiles for all evaluation options included into this study. The reason for the approximate comparison was the risk of the effect of the overall clotrimazole content on the obtained profiles.

## 4. Conclusions

A sequential injection system was used to automate the fast liberation of an active substance from semisolid pharmaceutical formulations. Clotrimazole served as the measured active substance in three tested formulations. Individual steps in the measurement process were optimized to keep the time needed for the determination of a single point as short as possible. The main aim was to discuss advantages and limitations of automated liberation testing using a fully automated system and relatively short sampling periods in the case of fast clotrimazole liberation. Detailed kinetic profiles measured in single and three liberation cells in parallel were used for the comparison of the tested formulations. The optimized measurement enabled comparison based both on liberation values obtained using shorter and longer sampling intervals (using full capacity of the flow system) and on the steady-state flux. The identified differences among the formulations were more pronounced in the detailed testing using short sampling periods in a single liberation cell testing, whereas longer periods did not enable reliable demonstration of fast release of clotrimazole. The results of our novel initial rate approach to liberation profiles evaluation correspond well with the results obtained in traditional evaluation methods. For fast dissolution/liberation profiles, such quick and simple evaluations can be applied for screening applications or to supplement the information obtained from these tests.

## Figures and Tables

**Figure 1 molecules-26-05538-f001:**
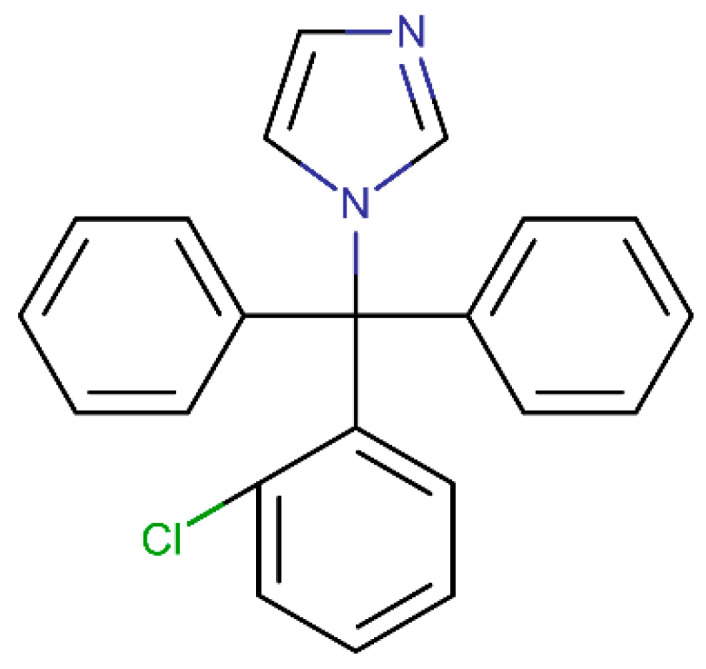
Chemical structure of clotrimazole [20].

**Figure 2 molecules-26-05538-f002:**
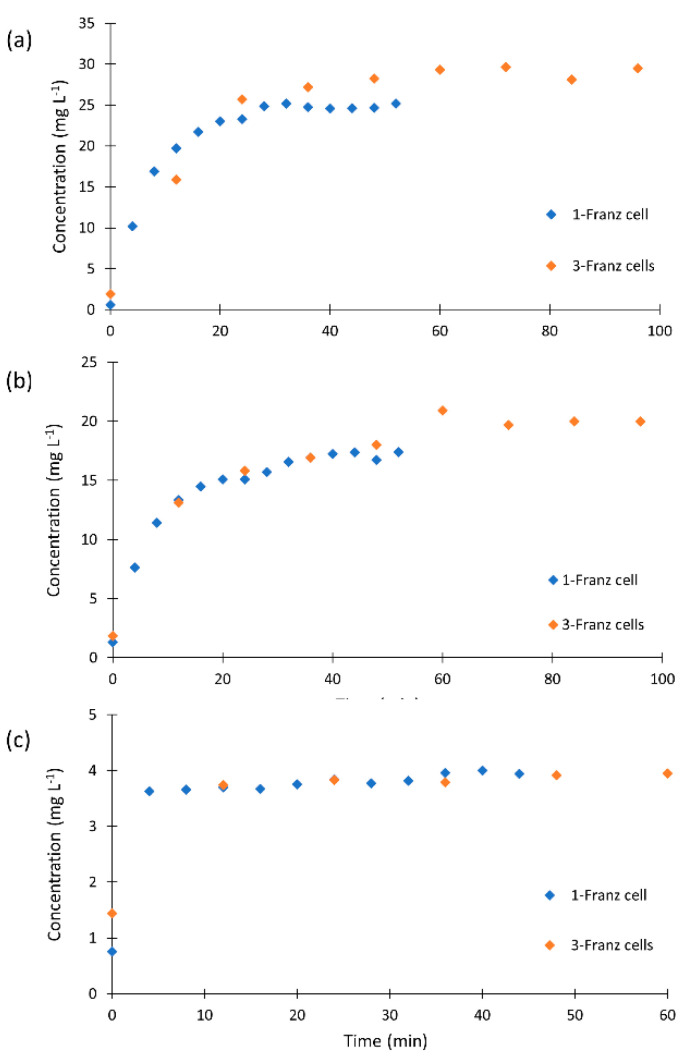
Comparison of the liberation profiles of (**a**) Canesten, (**b**) Clotrimazol AL, and (**c**) Delcore formulations obtained using one and three Franz cells.

**Figure 3 molecules-26-05538-f003:**
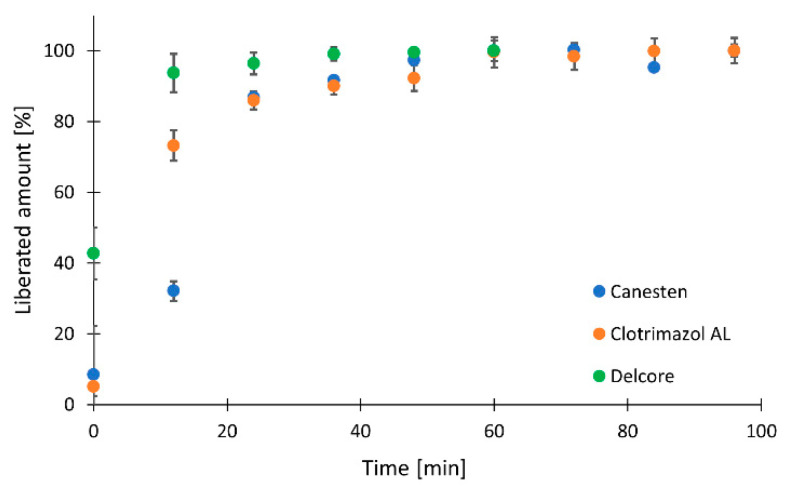
Clotrimazol liberated from Canesten, Clotrimazol AL, and Delcore formulations using three Franz cells in parallel.

**Figure 4 molecules-26-05538-f004:**
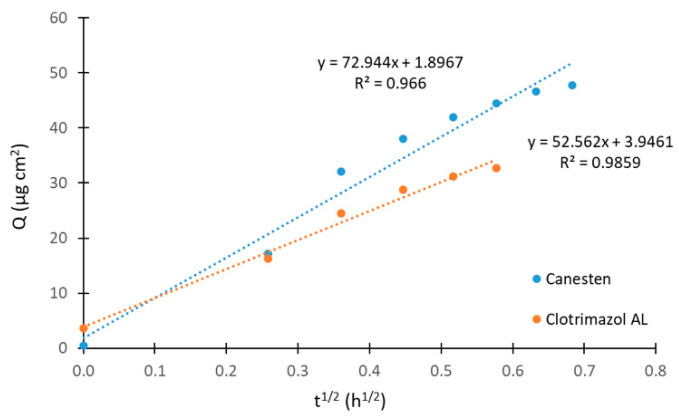
Linearization of liberated clotrimazole amount for Canesten and Clotrimazol AL formulations in the experiment with one Franz cell.

**Figure 5 molecules-26-05538-f005:**
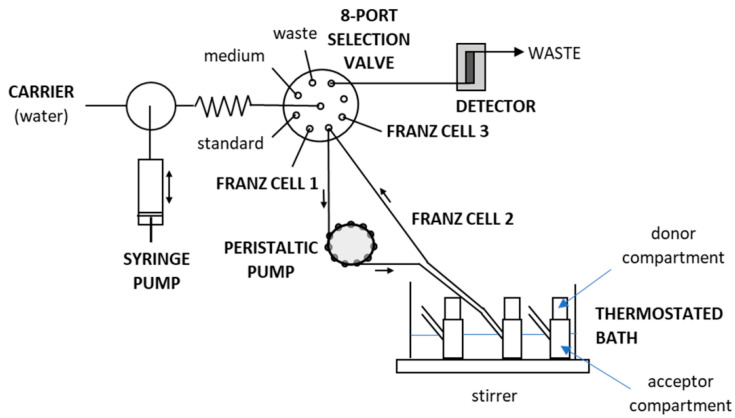
SIA system coupled to three Franz cells used for liberation testing.

## Data Availability

Data are contained within the article or Appendix A.

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
