# Peer review of "Sequential Injection Analysis for Automation and Evaluation of Drug Liberation Profiles: Clotrimazole Liberation Monitoring"

_molecules, 2021, doi:10.3390/molecules26185538_

Round 1
Reviewer 1 Report
This manuscript presents the limits of simple on-line determination using direct detection of clotrimazole. The effect of the sampling period length was discussed by critical comparison of profiles obtained with different sampling intervals. The authors propose a simple mode to calculate the initial liberation rate from the second and third sampling points of the registered profile to characterize the release of clotrimazole from three semisolid formulations. This approach is compared with the values obtained by the standard evaluation methods and discussed as an alternative to obtain the required information in shorter times on the example of different clotrimazole formulations with expected quick release of the active substance.
The experiments in this study are well planned and of good technical quality. However, several issues need to be addressed for the manuscript to be published in Molecules. To improve the manuscript I suggest the following comments:
- In the introduction, the authors must indicate other advantages of this method compared to other methods used to determine the release kinetics of drugs.
- It is not very well understood how much Delcore is used for testing. It is specified as: "The amount of clotrimazole creams used in the liberation test was 40 mg". In the following sentence the authors say that: "A Delcore dispersion of 300 µL was taken from 1 mL phosphate buffer where 5 mg of lyophilized was dissolved at the time before liberation test". Please also mention whether Delcore is in the form of a cream or not and how much clotrimazole is in the Delcore formulation used for the tests.
- if the same amount of clotrimazole was not used for the release tests in all 3 drug formulations, the comparison cannot be discussed.
Author Response
- In the introduction, the authors must indicate other advantages of this method compared to other methods used to determine the release kinetics of drugs.
The Introduction part was extended with the other advantages and other methods used for release monitoring were mentioned – mainly pharmacopeial requirements and bioequivalence studies were described. (lines 52-63)
- It is not very well understood how much Delcore is used for testing. It is specified as: "The amount of clotrimazole creams used in the liberation test was 40 mg". In the following sentence the authors say that: "A Delcore dispersion of 300 µL was taken from 1 mL phosphate buffer where 5 mg of lyophilized was dissolved at the time before liberation test". Please also mention whether Delcore is in the form of a cream or not and how much clotrimazole is in the Delcore formulation used for the tests.
Composition of all tested formulations including preparation of Delcore dispersion from lyophilizate were mentioned in the supplementary Table S1 and in the manuscript text. (lines 237-242, Table S1)
- if the same amount of clotrimazole was not used for the release tests in all 3 drug formulations, the comparison cannot be discussed.
The comparison of tested formulations was revised to be appropriate, and the authors are convinced that comparison based on percentage of liberated clotrimazole can be mentioned. But we agree that detailed comparison can be carried out only for Clotrimazol AL and Canesten where the same amount of clotrimazole in similar matrix (cream) is present. (lines 168, 173, 335-340)
Reviewer 2 Report
In this work, the authors investigated the development of a methodology for automation and evaluation of drug release profiles, using sequential injection analysis as strategy and clotrimazole as model drug (incorporated in three different pharmaceutical dosage forms). A sequential injection analyzer was applied to the automation of drug release studies. The authors concluded that the proposed method was suitable for the comparison of short-time kinetic profiles, and it can be used as a faster and simpler approach for dissolution/release testing. The work subject is very interesting; however, some problems and questions should be highlighted:
Q1. The authors must revise the abstract, mainly the first sentences, for clarifying the problem and the work’s aim.
Q2. Lines 34-47 – This paragraph of introduction should be revised.
Q3. Line 84 – Franz’s cell is a model of vertical diffusion cell frequently utilized for skin permeation studies; however, it can be used for other investigations such as drug release profile. This information must be emphasized in discussion.
Q4. Lines 102-108 – References must be added to the methodology, like ICH, 2005.
Q5. The authors used three different formulations and concentrations of clotrimazole. These differences can influence the results, and this must be more considered in discussion section.
Q6. More information about the composition of each formulation must be included and considered for discussion.
Q7. Conclusion should be improved considering the aim of the work and the results.
Q8. There is a low number of references, and some important (and up to date) ones must be included.
Author Response
Q1. The authors must revise the abstract, mainly the first sentences, for clarifying the problem and the work’s aim.
The recommended revisions were implemented into the abstract text. (lines 9-12)
Q2. Lines 34-47 – This paragraph of introduction should be revised.
The authors have revised Introduction with respect to other advantages of automated sampling in liberation studies, other applications of Franz cell design and the up-to-date articles were cited in this part. (lines 52-63)
Q3. Line 84 – Franz’s cell is a model of vertical diffusion cell frequently utilized for skin permeation studies; however, it can be used for other investigations such as drug release profile. This information must be emphasized in discussion.
Again, Franz cell applications were added to the Introduction chapter including more recent references. (lines 64-75)
Q4. Lines 102-108 – References must be added to the methodology, like ICH, 2005.
The reference to ICH was added to this part. (line 137)
Q5. The authors used three different formulations and concentrations of clotrimazole. These differences can influence the results, and this must be more considered in discussion section.
The comparison was revised concerning this comment, but the authors still would like to mention the possibility of comparison based on percentage of liberated clotrimazole. (lines 168, 173, 335-340)
Q6. More information about the composition of each formulation must be included and considered for discussion.
Detailed composition of all tested formulations was mentioned in the supplementary Table S1. (Table S1)
Q7. Conclusion should be improved considering the aim of the work and the results.
Conclusion was revised following this comment. (lines 346-348)
Q8. There is a low number of references, and some important (and up to date) ones must be included.
References were extended with articles focused on automation using Franz cells for liberation and permeation testing. (lines 64-75)
Round 2
Reviewer 1 Report
The authors paid attention to the requirements of the reviewers and modified the manuscript according to their suggestions